# Experimental Investigation on the Anode Flow Field Design for an Air-Cooled Open-Cathode Proton Exchange Membrane Fuel Cell

**DOI:** 10.3390/membranes12111069

**Published:** 2022-10-29

**Authors:** Zhijun Deng, Baozhu Li, Shuang Xing, Chen Zhao, Haijiang Wang

**Affiliations:** 1Research Institute of New Energy Vehicle Technology, Shenzhen Polytechnic, Shenzhen 518055, China; 2Internet of Things & Smart City Innovation Platform, Zhuhai Fudan Innovation Research Institute, Zhuhai 518057, China; 3Department of Mechanical and Energy Engineering, Southern University of Science and Technology, Shenzhen 518055, China

**Keywords:** proton exchange membrane fuel cell, anode, serpentine channel, depth, width, mass transfer

## Abstract

A flow channel structure design plays a significant role in an open-cathode proton exchange membrane fuel cell. The cell performance is sensitive to the structural parameters of the flow field, which mainly affects the heat and mass transfer between membrane electrode assembly and channel. This paper presents theoretical and experimental studies to investigate the impacts of anode flow field parameters (numbers of the serpentine channels, depths, and widths of the anode channel) on cell performance and temperature characteristics. The result indicates that the number of anode serpentine channels adjusts the pressure and flow rate of hydrogen in the anode flow channel effectively. The depth and width of the channel change the pressure, flow rate, and mass transfer capacity of hydrogen, especially under the high current density. There appears the best depth to achieve optimum cell performance. The velocity and concentration of hydrogen have important influences on the mass transfer which agrees with the anode channel structure design and performance changes based on the field synergy principle. This research has great significance for further understanding the relationship between anode flow field design and fuel cell performance in the open-cathode proton exchange membrane fuel cell stack.

## 1. Introduction

Nowadays, hydrogen, as a kind of new energy, has attracted increasing attention, and it is used to replace the decreasing fossil energy and solve increasingly serious environmental problems [1]. Proton exchange membrane fuel cell (PEMFC) technology has received considerable attention in the past two decades due to its high efficiency and low pollution [2,3,4,5,6]. PEMFC can be divided into the air-cooled PEMFC and liquid-cooled PEMFC according to different cooling methods. With the advantages of structural simplicity, lower weight, lower cost, and easy operation, the air-cooled open-cathode PEMFC is more favorable and applicable to the portable power source, unmanned aerial vehicle (UAV), and forklift [7,8,9].

The temperature and the mass transfer are both critical factors to influence the performance of open-cathode PEMFC [10]. Moreover, in the past few decades, many scholars have conducted research on the performance optimization of open-cathode PEMFC channels [11,12]. Jaeseung Lee et al. took the novel cathode flow field design for passive open-cathode PEMFC, which could enable the deceleration of the reactant air and acceleration of the coolant air simultaneously along the airflow path [13]. Sobi Thomas et al. analyzed the effect of the flow channel width and depth on pressure and velocity for HT-PEMFC [14]. Guo et al. put forward a bionics flow field, and combined simulation and experiment to study its effect on the velocity and pressure drop. The results show that the peak power density of the fuel cell using the interdigitated bio-inspired design is 20~25% higher than conventional designs. The ratio of cathode channel width to depth (0.83–2.5) was also investigated on an open-cathode fuel cell [15,16]. Diankai Qiu et al. conducted a numerical analysis with different cathode flow channels for open-cathode PEMFC; the results show that the cell performance is sensitive to the cathode channel structure, which affects the temperature distribution and mass transfer in the cells [17]. Afshari et al. set up the 3D cathode channels to obtain a better performance in air mass transfer and water removal by enhancing the fins [18]. Narissara et al. studied the effects of cathode flow-field designs (parallel slit, circular open, and oblique slit), and the results show that a circular opening design yields the best performance and the highest limiting current [19]. Jun Shen et al. reviewed the influence of the flow field based on the synergic, the results indicate that the synergic degree between the velocity vector and concentration gradient agrees with the performance changes, and the effective mass transfer coefficient is also enhanced [20]. Zhu et al. numerically studied the effects of channel geometry (rectangle, trapezoid, upside-down trapezoid, triangle et al.) on the evolution and motion of water droplets, flow resistance, water saturation, and coverage ratio [21]. It is found that with decreasing aspect ratio for the rectangle the pressure drop increases and the coverage ratio decreases. Moreover, a new approach for probing the operation of open-cathode fuel cells has been presented that uses a “hydro-electro-thermal” mapping process through the combined use of water imaging, current, and temperature mapping [22,23]. Park et al. designed a new serpentine flow field with sub-channels and by-passes to enhance the under-rib convection for water removal, based on which the maximum power densities increased by 23.74% [24]. Zhao et al. put forward that the cathode channel design parameters should be operated at appropriate width, as deep as possible, small ratio, and bending [25]. Moreover, as the channel size decreases, the total pressure drops across the cell increases, which leads to more pump work [26]. In addition, the channel optimized with a genetic algorithm, 3D fine mesh channel, and the channel with large length-width ratio are proposed for enhancement of cell performance [7,27,28,29].

Although many scholars have carried out a large number of theoretical and experimental studies on bipolar plate design parameters and operating conditions (flow rate, pressure, and temperature) for open-cathode PEMFCs, the design of the flow channel is more concentrated on the cathode. However, the effect of the anode channel design had not been investigated in the past. From the theoretical design of the flow channel for open-cathode PEMFC, the performance of the fuel cell is determined by the operating conditions (pressure, temperature, and humidity of the reaction gas) to meet the application requirements and constraints, and the pressure of the reaction gas determines the flow rate of the reaction gas in the flow channel. The cell performance is more sensitive to the flow rate of the reactant since the gas flow rate determines the mass transfer effect and field synergy effect in GDL. The design theory applies not only to the cathode but also to the anode.

In this work, we present the different anode model, the number of serpentine flow channels, and the depth and width of flow channels were set as structural variables to investigate the influence of these structural design parameters on fuel cell performance. The experimental results showed that the different structure design parameters of the anode channel also have an obvious influence on the performance. Furthermore, the purpose of this study is to obtain basic data on the interdependence of these anode channel design parameters and their impact on cell performance. Such a foundation was necessary for optimizing the flow channel design of open-cathode PEMFC.

## 2. Theory Design

For open-cathode PEMFC, it has the characteristics of low hydrogen pressure and normal temperature at the input end. Simultaneously, the hydrogen flow rate should be as small as possible to meet the normal operation of the fuel cell, since open-cathode PEMFC is different from liquid-cooled PEMFC, and there is no hydrogen circulating pump, and the hydrogen is discharged periodically. This characteristic requires optimization design of the anode channel, which enables the fuel cell to achieve a better hydrogen mass transfer effect and electrical performance.

The movement process of hydrogen in the flow channel is a typical convective mass and heat transfer process, which conforms to the ideal gas state equation, Fourier heat conduction law, and Fick diffusion law [30]. The expressions are as follows:
*PV* = *nRT*(1)
(2)$$q_{h}\;=-k\frac{dT}{dn}$$
(3)$$q_{m}=-\rho D\frac{dY}{dn}$$
where P is the pressure, Pa; *V* is the volume, m^3^; *n* is the molar, mole; *R* is the gas constant, 8.314 J/(mol·K); T is the temperature, K; $q_{h}$ is the heat flux, J/m^2^; *k* is the thermal conductivity, W/(m·K); $\frac{dT}{dn}$ is the rate of temperature change perpendicular to the interface, K/m; ρ is the density, kg/m^3^; $q_{m}$ is the diffusion flux, kg/(m^2^·s); D is the diffusion coefficient, m^2^/s; $\frac{dY}{dn}$ the concentration gradient, kg/m^4^.

From Equations (2) and (3), the transport capacity is directly proportional to the gradient of a corresponding physical quantity in the process of heat and mass transfer, i.e., the value of mass transfer has a linear relationship with a certain generalized force. For the hydrogen input of open-cathode PEMFC, the flow rate was set at 0.5 L/min and the pressure was set at 50 kPa, the flow state belongs to laminar flow. According to the field synergy theory in the laminar mass transfer process, the component concentration equation of the steady-state laminar convective mass transfer process without component sources is [31,32]:(4)ρU·∇Y=∇·(ρD∇Y)
where ρ is the density; *U* is the velocity; ∇Y is the component concentration gradient.

In the whole convective mass transfer region Ω, Equation (4) is integrated, then the volume integral is transformed into the area integral on the boundary by Gauss, the result is given as:(5)$$\int_{\Omega }\nabla \cdot \left(\rho D\nabla Y\right)dV=\int_{\delta }n\cdot \left(\rho D\nabla Y\right)dS$$

For the convective mass transfer processes in most engineering fields, the Peclet number is all greater than 100. Moreover, when the Peclet number is greater than 100, the component diffusion in the fluid is negligible relative to flow mass transfer. So, Equation (5) can be expressed as Equation (6):(6)$$\int_{\Omega }\rho U\cdot \nabla YdV=\int_{ms}n\cdot \left(\rho D\nabla Y\right)dS$$

It can be seen from Equation (6) that the integral of the point product of velocity and component concentration gradient in the whole convective mass transfer region is equal to the convective mass transfer mass of the whole system. Here, the equivalent characteristic length is defined as L, and the expression is shown in Equation (7). The following dimensionless variables are introduced, see Equation (8).
(7)$$L=\frac{V}{S}$$
(8)$$\bar{U}=\frac{U}{U_{in}},\;\nabla \bar{Y}=\frac{\nabla Y}{\frac{Y_{ms}-Y_{in}}{L}},\;d\bar{V}=\frac{dV}{V}$$
where *V* is the volume of the convective mass transfer region; *S* is the area of the mass transfer surface.

The dimensionless variables in Equation (8) are introduced into Equation (6), and the dimensionless relationship of the convective mass transfer process is obtained as follows:(9)$$\mathrm{Sh}=\mathrm{ReSc}\int_{\Omega }\bar{U}\cdot \nabla \bar{Y}d\bar{V}$$
where $\mathrm{Sh}=\frac{h_{m}d}{D}$ is the Sherwood number; Sc is the Schmidt number; $h_{m}$ is the convective mass transfer coefficient.

It can be seen from Equation (9) that the effect of convective mass transfer depends not only on the Re and Sc but also on the integral of $\bar{U}\cdot \nabla \bar{Y}$ in the whole region, see Equation (10):(10)$$F_{C_{m}}=\int_{\Omega }\bar{U}\cdot \nabla \bar{Y}d\bar{V}$$

For the convective mass transfer process of laminar, the Schmidt number of the flow is generally a fixed value. Thus, to enhance the convective mass transfer process, the Reynolds number can be increased, i.e., the local velocity of the flow can be increased; or the synergy of the mass transfer field can be increased, i.e., the velocity field and concentration field of the flow can be changed by changing the geometric structure parameters of the boundary of the convective mass transfer region.

The pressure, flow rate, and stoichiometric ratio of hydrogen are kept at a certain level in the inlet position, and the hydrogen gas flow and the channel plane flow present a horizontal direction, while the electrochemical reaction of the fuel cell occurs in the vertical direction. Simultaneously, the transfer and reaction of hydrogen and oxygen in the membrane electrode assembly (MEA) is a convective mass transfer process, the convective of hydrogen is caused by the GDL which is the porous media. If the horizontal flow rate of hydrogen is too high, the efficiency will be reduced (because hydrogen will be wasted and carries away moisture). Then if the flow rate of hydrogen is too low, the amount of hydrogen entering MEA is not enough, and the cell performance will be limited. However, by changing the structural design parameters of the hydrogen channel, the pressure, velocity, and flow path of hydrogen in the channel can be changed, the convective mass transfer capacity of hydrogen also can be changed, and the reaction rate can be changed, so that hydrogen can form local turbulent flow even at low Reynolds number. Based on this theory, the theoretical calculation and performance test was carried out by changing the numbers of anode serpentine channels, depths, and widths of the anode channel as shown in Figure 1 and Table 1.

For the designed hydrogen flow field with different parameters above, the pressure drop is different. According to Equations (11)~(14), the pressure drop of different anode structural design parameters can be obtained, as shown in Figure 2.
(11)$$\Delta P=f\frac{L}{D_{H}}\rho \frac{{\bar{v}}^{2}}{2}+\sum K_{L}\rho \frac{{\bar{v}}^{2}}{2}$$
(12)$$D_{H}=\frac{2w_{c}d_{c}}{w_{c}+d_{c}}$$
(13)Re=ρv¯DHμ
where $D_{H}$ is the hydraulic diameter, m; *L* is the channel length, m; *v* is the velocity of gas flow, m/s; *f* is the channel friction factor; $K_{L}$ is the local impedance; $w_{c}$ is the channel width, m; $d_{c}$ is the channel depth, m; μ is the viscosity of the fluid, kg/(m·s).

For the static laminar flow in the channel, the product of friction factor and Reynolds number is constant $R_{e}f$, for rectangular channels:(14)$$R_{e}f\approx 55+41.5\exp\left(\frac{-3.4}{w_{c}/d_{c}}\right)$$

For the coefficient of local impedance, although some geometric pressure loss coefficients can be used for all kinds of elbows, there is not a gas flow channel suitable for specific shapes in the fuel cell. Thus, for 90° elbow in the fuel cell, 30f is generally selected for theoretical calculation and design.

Figure 2 shows the comparison results of pressure drop and channel length under different anode channel structure parameters. The anode pressure drop decreases significantly with the increase in channel number (Figure 2a), and the six-serpentine channel has the smallest pressure drop (13.5 kPa). Moreover, under the same number of serpentine channels, the anode channel with higher depth and width could achieve a smaller pressure drop. The more the number of anode serpentine channels, the smaller the total length of the anode channel (Figure 2b). Simultaneously, the channel length decreases with the increase in the channel width. According to Figure 2 and Equation (11), both the flow rate and pressure drop of hydrogen are closely related to the length of the flow channel. According to the field synergy theory, the change in the flow rate could affect the field synergy factor of mass transfer, and then affect the concentration of hydrogen entering MEA. Therefore, through theoretical calculation, it could be concluded that the anode flow field design parameters had great significance in the open-cathode PEMFCs, which was further verified by the following experiments.

In the fuel cell, it involves the interaction of multiple physical fields, mainly including velocity, pressure, temperature, matter, current, voltage, and other physical fields. However, in an open-cathode fuel cell, the heat dissipation of the cell is coupled with the oxygen supply, which further deepens the coupling of the physical fields. Therefore, sorting out the synergy between the physical fields is conducive to a better and more accurate understanding of the design of the channel with high performance.

## 3. Experimental

### 3.1. Experimental Setup

A single open-cathode PEMFC (50 cm^2^) was designed and fabricated in-house as depicted in Table 1 and Figure 1. The anode and cathode flow fields are fabricated with serpentine and parallel channels respectively. Commercial catalyst-coated membrane (CCM) and GDL were employed in the assembly of MEA. The thickness of CMM is 29 μm, the anode and cathode Pt loadings are 0.1 mg/cm^2^ and 0.4 mg/cm^2^, respectively. Table 2 lists the detailed parameters of CCM and GDL. 

The electrochemical test was carried out on the 100 W fuel cell test bench (850e, Hephas, Taiwan), which includes the power measurement subsystem, fuel supply subsystem, gas humidification subsystem, temperature control subsystem, and data recording subsystem. The maximum testing voltage and current of the electrical load of the electrochemical test device are 20 V and 100 A, respectively. Moreover, the flow rate ranges of H_2_ and air are 0~1 SLPM and 0~2 SLPM, respectively.

The San Ace (90) tube-axial fan with blowing mode was installed at the outside of the cathode. The function of the fan was to provide enough oxygen for the chemical reaction in the fuel cell and remove the waste heat generated by the cell promptly to ensure the cell temperature was within the suitable operating temperature range avoiding water flooding and dehydration of MEA.

### 3.2. Experimental Conditions and Procedure

The temperature and relative humidity (RH) of ambient air were between 20~26 °C and 30~45%, respectively. Dry and non-heated hydrogen (purity 99.99%) was supplied to the anode of PEMFCs. The cathode outlet temperature distribution was measured to quantify the effect of different anode flow field designs. The thermography had been made with a TESTO^®^ 865 camera (temperature range: −20~280 °C), with a matrix of 320 × 240 sensors and a thermal resolution (NETD) < 120 mK.

Moreover, it is noteworthy that the MEA needs to be activated before the parameter experiment. During the activation process, the discharge current is gradually adjusted from the open-circuit state to 1.0 A/cm^2^ and kept constant until the cell voltage is stable (the voltage fluctuation is less than 5 mV). The activation process usually lasts for 4~6 h according to the different original states of MEA before activation. For polarization curves and power plots, the constant current mode is applied (0~1.0 A/cm^2^), 0.1 A/cm^2^ per point, and each point sustains 2 min. Simultaneously, HFR data are obtained through the test equipment. The electrochemical impedance spectroscopy (EIS) of the open-cathode fuel cell is measured at 0.8 A/cm^2^. The sinusoidal disturbance of current (i.e., alternating current, AC) is added to direct current (DC), and the voltage response is obtained. The AC frequency is scanned from 10 kHz to 0.1 Hz. The amplitude of AC is 10% of DC. 

Each experiment was conducted three times to reduce the influence of accidental factors on the experiment and take the average values as the final results. The detailed open-cathode fuel cell experimental parameters were listed in Table 3.

## 4. Results and Discussion

In this section, the results of the polarization curve, EIS, temperature distribution, and stable output response of open-cathode PEMFCs with different anode channel parameters are discussed in detail.

The structure design of the anode channel is important due to its role in the mass transfer of hydrogen. Based on the theoretical calculation, to explore the effect of anode channel design parameters, some single fuel cells with commercial MEA were used to experimental analyze the influence of different serpentine numbers (2, 3, 4, 6), channel depths (0.2, 0.3, 0.4, and 0.5 mm), and channel widths (0.35, 0.45, 0.55, and 0.65 mm). In the process, the basic design anode channel parameters were four serpentine channels with a depth of 0.3 mm and width of 0.45 mm, while the ratio of channel width to landing width was 1:1. In the experiment, when one of the parameters changes, the other two parameters remain unchanged.

### 4.1. Effect of Anode Serpentine Channel Number

The polarization curves of the cell with different numbers of anode serpentine channels were compared in Figure 3. From Figure 3a, the PEMFC performance increases when the number of serpentine channels increases from 2 to 6. This phenomenon is more obvious with the increase in current density. The performance of the cell with a six-serpentine channel is 10.7% higher than that of the cell with a two-serpentine channel at 0.8 A/cm^2^ since the cell voltages are 0.563, 0.568, and 0.573 V when the number of serpentine channels is 3, 4, and 6. When the number of serpentine channels is greater than or equal to 3, although the cell performance still shows an increasing trend with the increase in the number of serpentine channels, the absolute value of performance increase decreases and remains at about 5 mV. With the increase in the number of serpentine channels, the pressure drop decreases, the overall gas pressure in the cell increases, and the hydrogen mass transfer resistance decreases.

Although the EIS test method has obvious cathode characteristics, due to the change of the structure and size in the anode side channel for fuel cell bipolar plate, which has different effects on the flow pressure and mass transfer for hydrogen into MEA. Therefore, the EIS test is still a good way to characterize the loss in the above test cases. EIS can characterize and analyze the ohmic, charge transfer, and mass transfer resistance, which is the most commonly used research fuel cell technology. To evaluate and compare the impedances of the fuel cell with different anode channel design parameters, EIS tests were conducted at 0.8 A/cm^2^. 

The EIS of open-cathode PEMFC with different anode channel design parameters (numbers of the serpentine channels, depths, and widths) were shown in Figure 3b. The anode channel design parameters have little influence on the ohmic resistance since the contact resistance mainly depends on the opening rate of the bipolar plate and the contact area between the bipolar plate and GDL. However, with the change in the number and depth of anode serpentine flow channels, the fluctuation range of the opening rate is 50.25~50.44% and 50.08~50.61%, respectively, showing no obvious change. See Figure 3 for details. Therefore, the ohmic resistance of different anode design parameters is basically in the range of 0.105~0.115 Ω cm^2^. Moreover, the value of total polarization resistance decrease with the increase in the number of anode serpentine channel. When the number of serpentine channels reaches 6, the value of total polarization resistance is 0.225 Ω cm^2^ which is the smallest. While the difference in total polarization resistance between the six-serpentine channel and the two-serpentine channel is 0.02 Ω cm^2^. Since the number of runner cycles and the length of hydrogen serpentine channels decrease and the pressure drop decreases from inlet to outlet under the same reaction area with the increase in the number of serpentine flow channels. Therefore, it leads to an increase in anode pressure and a more uniform distribution of hydrogen.

For open-cathode fuel cells, the operating temperature is an essential factor. Although the cooling capacity of the open-cathode fuel cell is mainly determined by the cooling capacity of the airflow on the cathode side, the performance and pressure are also affected by the structure of the anode serpentine channel according to Figure 2 and Figure 3. Thus, to confirm the influence of hydrogen channel structure parameters on the temperature distribution of open-cathode fuel cells, the temperature distribution of the cathode side outlet surface was continuously measured at 0.8 A/cm^2^.

Figure 4 shows the detailed temperature distribution for the cell with different anode serpentine channel numbers. The cathode outlet temperature presents volcanic distribution, with the hot spot temperature appearing in the middle, and the temperature decreases from the middle to both sides. Moreover, Figure 4b,c show that the hot spot temperature difference between the six-serpentine channel and two-serpentine channel is about 2.2 °C, the average temperature difference is about 0.6 °C, and the minimum temperature difference is 1.8 °C. The maximum, minimum, and average temperatures gradually decrease with the increase in the number of the serpentine channel as seen in Figure 4b. Since with the increase of serpentine channel number, the length of the corresponding single-channel becomes smaller, the pressure drop decreases, the hydrogen pressure in the channel increases, and the concentration of hydrogen increases, which makes the electrochemical reaction more active, resulting in enhanced anode heat dissipation capacity and electric performance, and reduced waste heat. Thus, the cathode outlet surface temperature decreases. 

For the open-cathode PEMFC, the load variation is another important factor affecting the temperature distribution of fuel cells. Thus, the influence of current densities on the cathode surface temperature under different anode channel structure design parameters is compared and analyzed. The test process is as follows: the current density gradually increases from 0 to 0.8 A/cm^2^, 0.1 A/cm^2^ is taken as a step, and the data are recorded after running for about 2 min under each current load.

Figure 4c shows the temperature curves under different currents for different anode serpentine channel numbers. The hot spot temperature is strongly correlated with the current load. The larger the current load, the greater the overpotential required, the more heat is generated, and the higher the temperature is. Moreover, the hot spot temperature is close for the cell with different anode serpentine channel numbers when the current load is less than 0.4 A/cm^2^. For the different number of serpentine channels, the temperature difference is within 0.5 °C. When the current density was greater than 0.4 A/cm^2^, the hot spot temperature shows an obvious difference, the hot spot temperature of the two-serpentine channel showed an obvious upward trend, and the gradient increased gradually. However, the hot spot temperature rising trend of the six-serpentine structure is slower than that of the other three serpentine channel numbers. Multi-serpentine number channels with lower flow resistance could effectively improve heat dissipation. Moreover, the cell with multi-serpentine number channels achieves high voltage at the same current density, which means that a lower overpotential is required, and the heat generated is lower. It is also beneficial for the decrease of cell temperature.

### 4.2. Effect of Anode Serpentine Channel Depth

The anode channel size of open-cathode PEMFC affects the pressure drop of the reaction gas and heat transfer, and thus the cell performance. Figure 5 shows the polarization curve and power density curve of the cell with different anode channel depths. With the increase of anode channel depth, the cell performance first increases and then decreases. Firstly, with the increase of channel depth, the flow resistance decreases, the pressure drop decreases, and the flow rate increases, which is conducive to enhancing the mass transfer of hydrogen from the channel to GDL. However, the channel depth continues to increase, and the excessive hydrogen flow rate carries a lot of moisture, resulting in the increase of ohmic impedance and the decline of cell performance, which is unfavorable to the open-cathode PEMFC without hydrogen humidification operation. When the anode channel depth is 0.4 mm, the fuel cell reaches a maximum power output of 456.0 mW/cm^2^ at 0.8 A/cm^2^, which is 6.49% and 9.12% higher than the cell with an anode channel depth of 0.2 mm and 0.5 mm, respectively. However, there appears a slight gap (1.6 mW/cm^2^) compared with that of the cell with the anode channel depth of 0.3 mm. The increase in channel depth would reduce the mechanical strength of the bipolar plate. Therefore, considering the mechanical strength of the bipolar plate and cell performance, it is more appropriate to select a 0.3 mm anode channel depth.

Moreover, Figure 5b shows the EIS of the open-cathode PEMFC with different anode channel depths. The value of total polarization resistance for the cell with an anode channel depth of 0.5 mm was the biggest, it is 0.03 Ω cm^2^ higher than the cell with an anode channel depth of 0.2 mm. The increase of total polarization impedance is mainly due to the increase of hydrogen flow, which takes away a large amount of water, resulting in the drying of the anode membrane and the decrease in the proton transport rate.

Similarly, Figure 6 reflects the cathode outlet surface temperature distribution for the cell with different anode channel depths. The hot spot temperature and the average temperature of the outlet surface first decrease and then increase as the channel depth increases. With the increase of channel depth, the flow resistance decreases, the flow rate of hydrogen is increased, and the heat dissipation capacity of the anode side is enhanced, so the cell temperature decreases. However, as the depth continues to increase, the faster flow rate of hydrogen takes away a large amount of water, resulting in membrane dehydration, increased cell resistance, and waste heat; thus, the cell temperature increases. It can be concluded that the negative effect of waste heat increase exceeds the positive effect of velocity increase on heat dissipation.

Moreover, the temperature rise rate of the anode channel with a depth of 0.2 mm and 0.5 mm is significantly faster than that of the other two channels (Figure 6c). When the channel depths are 0.3 and 0.4 mm, the temperature difference between the hot spot temperature and the minimum temperature is only 3.2 °C; there appears to be better synergy, compared with 7.5 °C when the depth is 0.5 mm. That means for the design of an anode channel, there is an optimal depth range to achieve an optimal matching relationship between performance and temperature. 

### 4.3. Effect of Anode Serpentine Channel Width

Then, the anode channel width is varied to improve cell performance with anode channel depth fixed at 0.3 mm and serpentine anode channel number fixed at 4. Figure 7 shows the performance of the cell with different anode channel widths. With the increase in width (0.35~0.65 mm), the cell performance increases. The increase in channel width enhances the hydrogen mass transfer capacity of the cell, but the contact area with the bipolar plate is reduced, which increases the contact resistance between GDL and the bipolar plate. From the perspective of the cell polarization curve, under the explored anode channel widths, the positive impact of enhanced mass transfer on cell performance is greater than the negative impact of increased ohmic resistance.

It also can be seen from Figure 7b that the value of total polarization resistance is decreasing with the increase in the width of the anode channel. However, when the width is 0.65 mm, the performance is the best, and the total polarization resistance is the smallest, which reaches 0.215 Ω cm^2^. Moreover, the whole electrochemical impedance trend exactly matches the trend of the performance curve.

For different channel widths, from Figure 8, the hot spot temperature with a channel width of 0.35 mm increases significantly when the current load is greater than 0.4 A/cm^2^ compared with the other three channel widths. The hot spot temperature growth trend of 0.65 mm channel width is the slowest. The difference between hot spot temperature and minimum temperature decreases with the increase of channel width, and when the channel width is 0.65 mm, the difference between hot spot temperature and minimum temperature is only 2.2 °C, which is the smallest among the four different channel widths.

The hot spot temperature and the average temperature of decrease as the channel width increases in the range from 0.35 mm to 0.65 mm. Moreover, when the width of the anode channel increases to a certain value, the temperature difference value also gradually decreases. Since as the channel width increases, the number of anode channels and the length of the channel also decreases, resulting in the increase of pressure in the channel, the improvement of fuel cell performance, the corresponding waste heat reduction, and the temperature of the cathode side outlet surface decrease. 

## 5. Conclusions

In this paper, the operating characteristics of open-cathode PEMFC with different anode flow field design parameters were verified by experiments. For the open-cathode PEMFC, which is highly coupled with oxygen mass transfer and heat dissipation, we analyzed the synergy between the physical fields, such as velocity, pressure, matter, and temperature based on the field synergy theory to guide the design of anode flow field. The specific conclusions are as follows:

The design parameters of the anode channel structure have an important impact on the performance of the fuel cell, which can change the interaction between various physical fields in the cell. Increasing the number of anode serpentine channels can effectively reduce the voltage drop, improve the mass transfer of hydrogen and heat dissipation, and improve the performance of fuel cells, especially under high current density. In addition, when increasing the depth of the anode channel, it is necessary to balance the negative effects of increased heat source and the positive effects of increased flow rate, and there is an optimal anode channel depth to achieve optimal cell performance. The width of the anode channel has relatively little effect on the performance, while the larger the width, the smaller the hydrogen pressure drop, and the greater the flow rate.

The performance of open-cathode PEMFC is affected by multiple factors. In addition to the flow channel design, the hydrophobic and heat dissipation enhancement design of MEA materials is also a major research focus to improve cell performance.

## Figures and Tables

**Figure 1 membranes-12-01069-f001:**
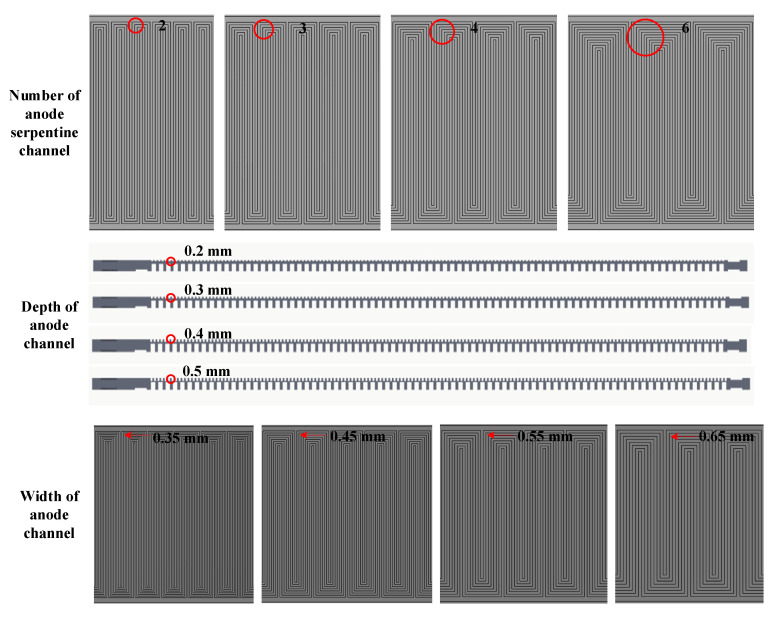
Schematic of anode flow field design with different channel numbers, depths, and widths of the anode serpentine flow field.

**Figure 2 membranes-12-01069-f002:**
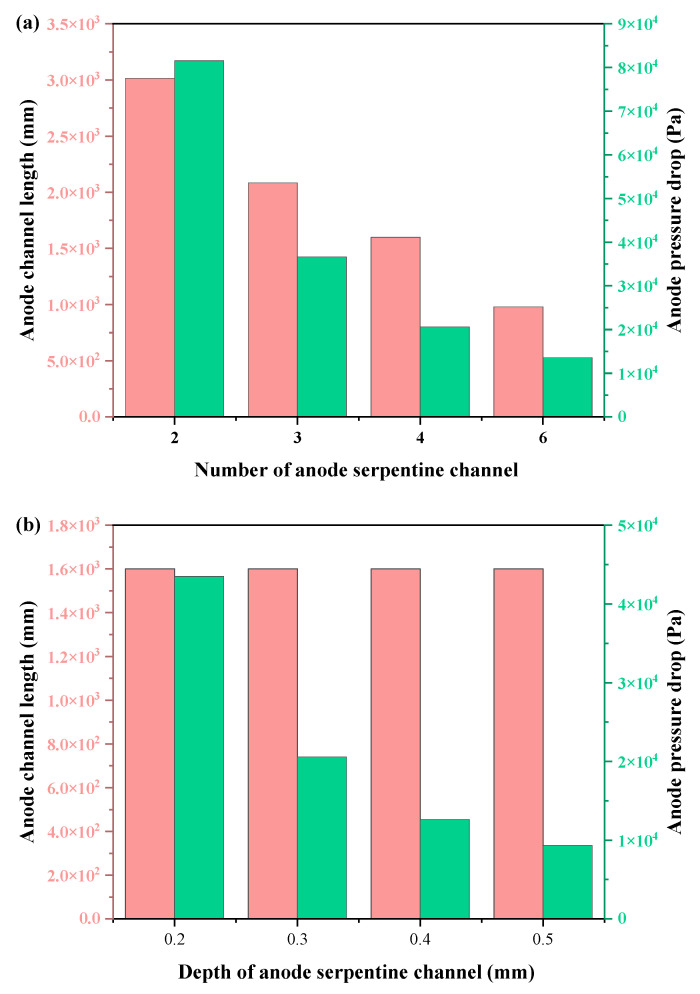
Calculation results of anode pressure drop and channel length of the cell with different anode serpentine channel parameters: (**a**) number, (**b**) depth, and (**c**) width.

**Figure 3 membranes-12-01069-f003:**
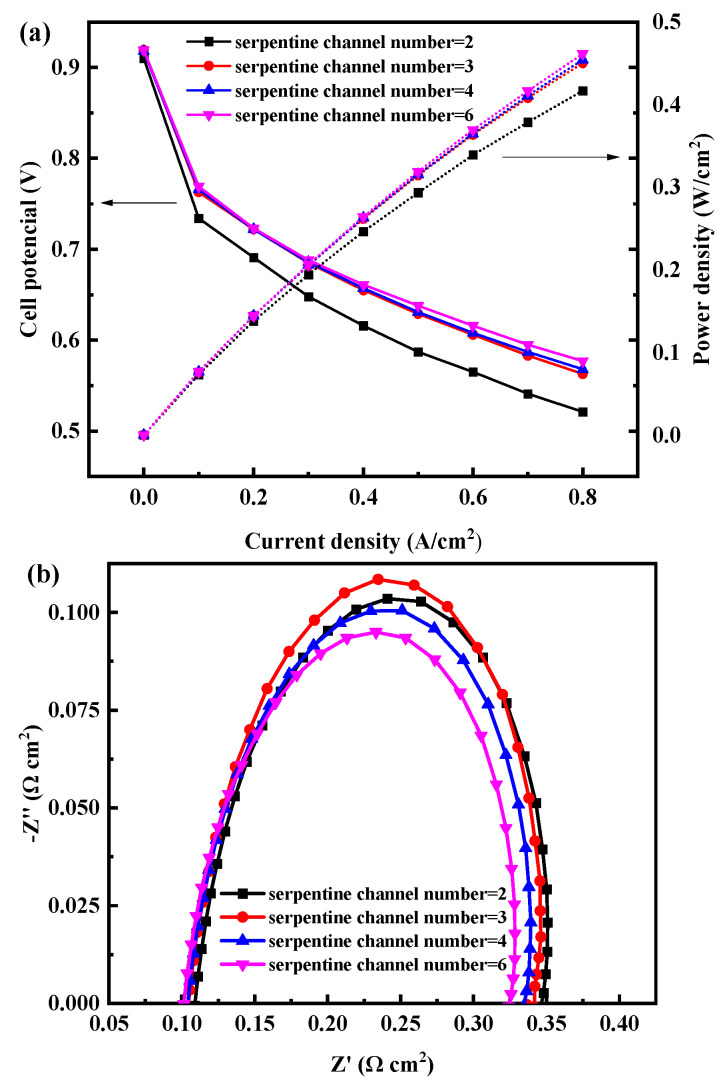
(**a**) Polarization curves and (**b**) power density curves of the open-cathode PEMFC with different anode serpentine channel numbers.

**Figure 4 membranes-12-01069-f004:**
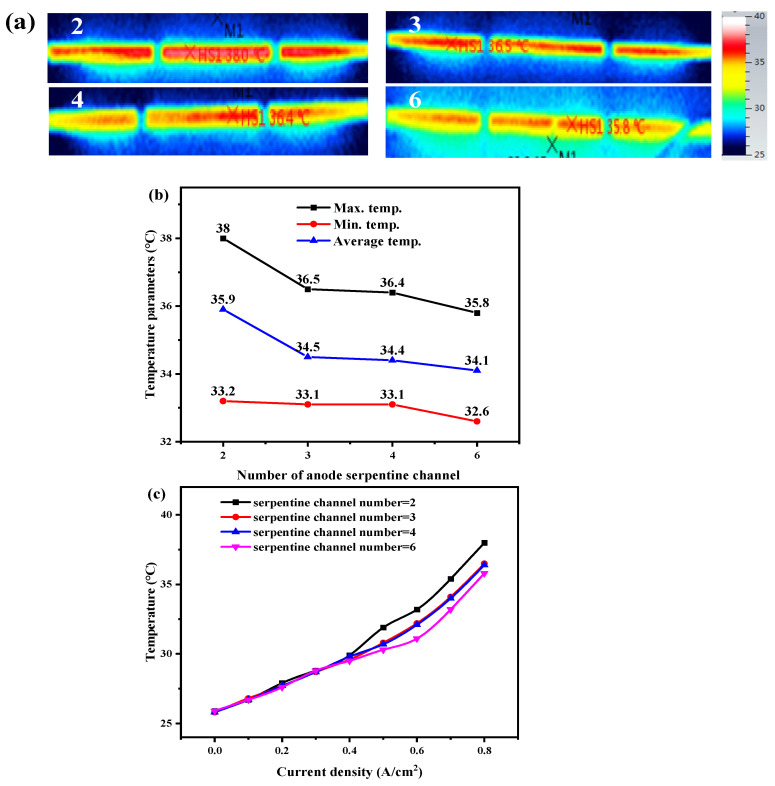
(**a**) Temperature distribution at 0.8 A/cm^2^, (**b**) maximum, minimum, and average temperatures, and (**c**) variation of hot spot temperature with current densities of the open-cathode PEMFC with different anode serpentine channel numbers.

**Figure 5 membranes-12-01069-f005:**
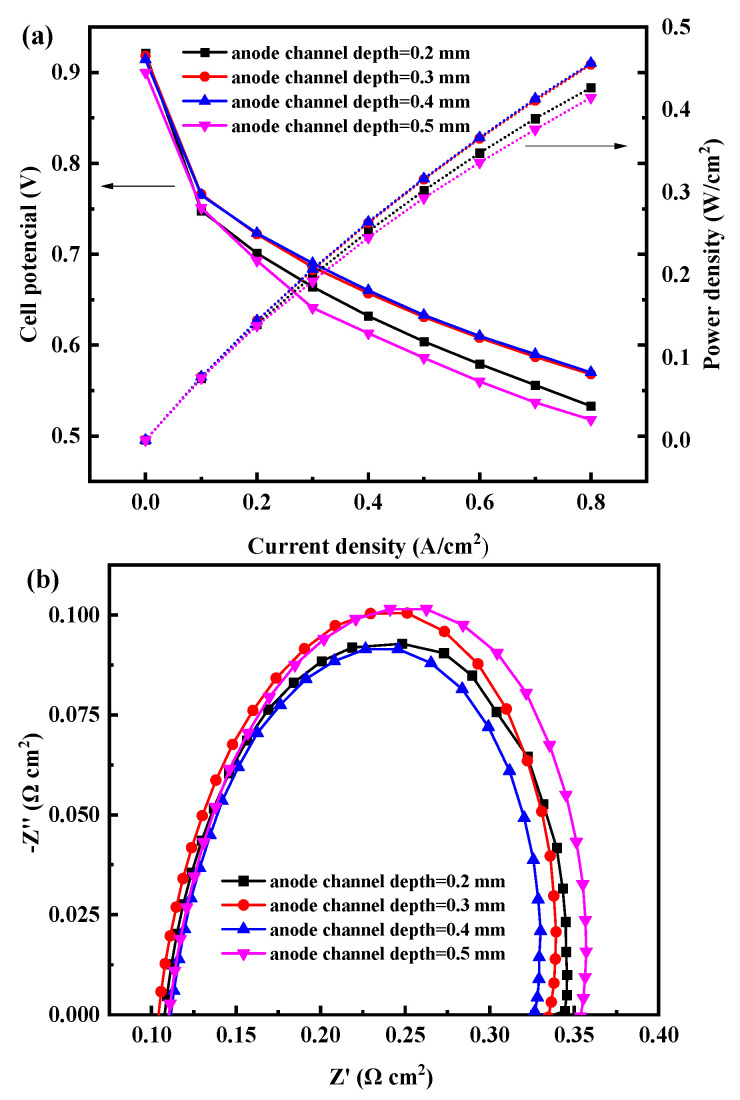
(**a**) Polarization and (**b**) power density curves of the open-cathode PEMFC with different anode serpentine channel depths.

**Figure 6 membranes-12-01069-f006:**
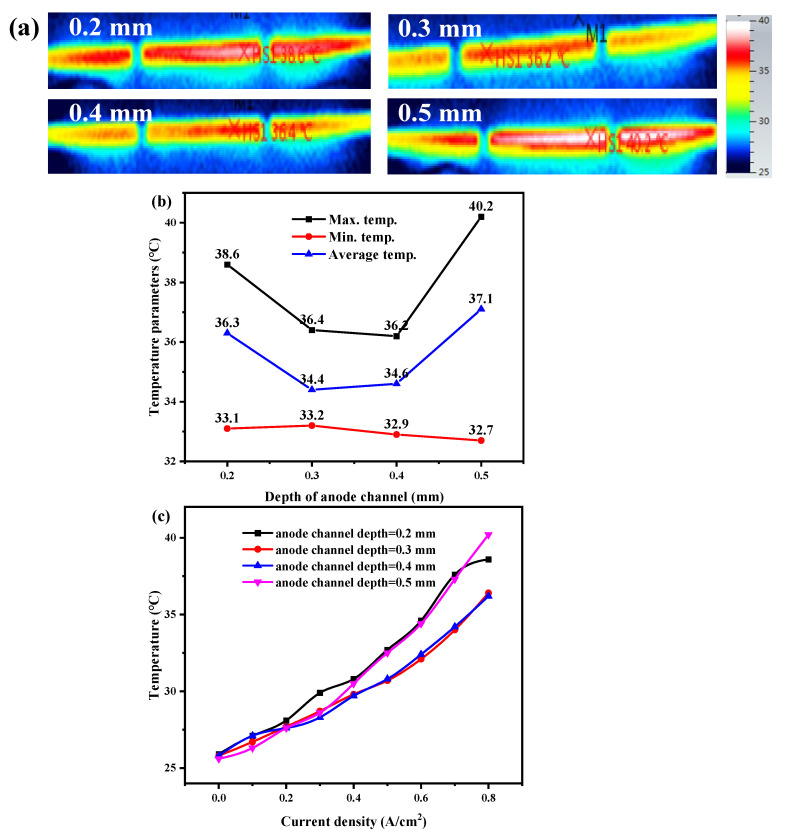
(**a**) Temperature distribution at 0.8 A/cm^2^, (**b**) maximum, minimum, and average temperatures, and (**c**) variation of hot spot temperature with current densities of the open-cathode PEMFC with different anode serpentine channel depths.

**Figure 7 membranes-12-01069-f007:**
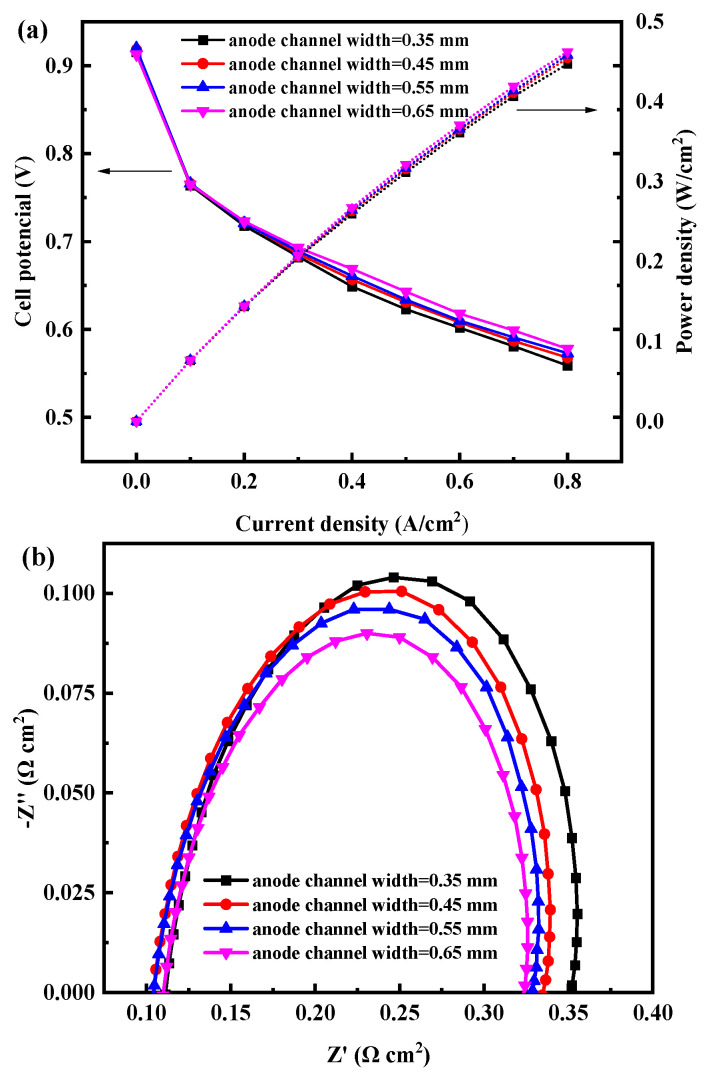
(**a**) Polarization and (**b**) power density curves of the open-cathode PEMFC with different anode serpentine channel widths.

**Figure 8 membranes-12-01069-f008:**
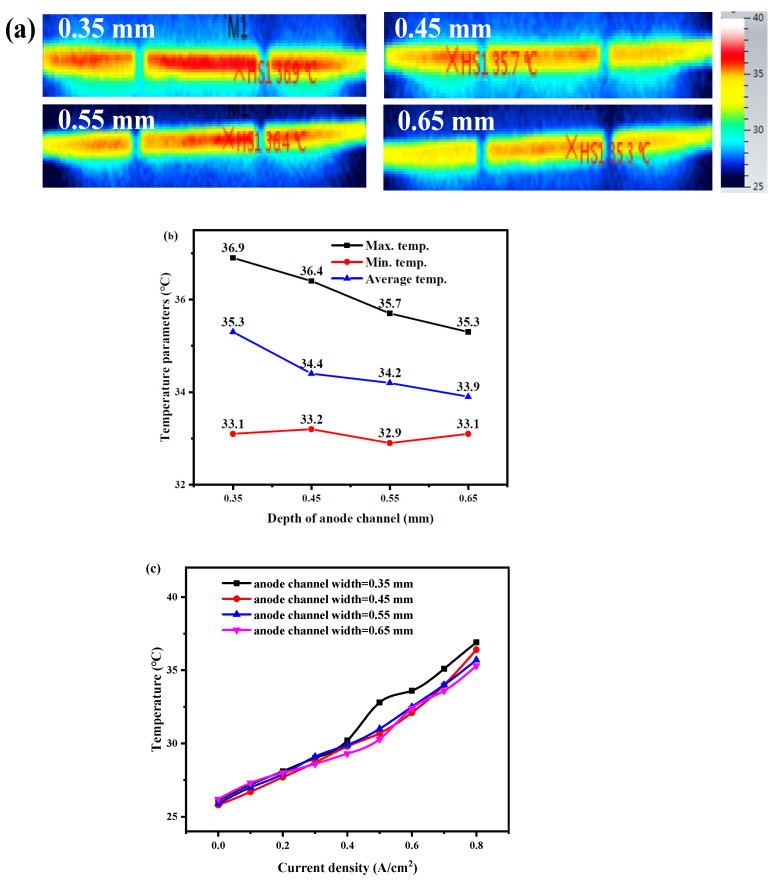
(**a**) Temperature distribution at 0.8 A/cm^2^, (**b**) maximum, minimum, and average temperatures, and (**c**) variation of hot spot temperature with current densities of the open-cathode PEMFC with different anode serpentine channel widths.

**Table 1 membranes-12-01069-t001:** Parameters of different anode serpentine flow fields for open-cathode PEMFC.

Parameter	Value
Channel number	2, 3, 4, 6
Depth of anode channel	0.2 mm, 0.3 mm, 0.4 mm, 0.5 mm
Width of anode channel	0.35 mm, 0.45 mm, 0.55 mm, 0.65 mm

**Table 2 membranes-12-01069-t002:** Parameters of the material of open-cathode PEMFCs.

Parameter	Value
CCM thickness	29 μm
Anode Pt loading	0.1 mg/cm^2^
Cathode Pt loading	0.4 mg/cm^2^
GDL thickness	200 μm
GDL area weight	55 g/m^2^
GDL electrical resistivity	8 mΩ/cm^2^
GDL density	0.27 g/ccm

**Table 3 membranes-12-01069-t003:** Experimental parameters of open-cathode PEMFCs.

Parameters	Definition	Value
V_cell_	Cell voltage	0.3~1.0 V
i_cell_	Cell current density	0~1.0 A/cm^2^
P_cell_	Cell power	0~23.5 W
Q_cell_	Heat generated by cell	0~46.2 W
T_H2_	Temperature of inlet hydrogen	30 °C
T_Air_	Temperature of inlet air	20∼26 °C
λ_H2_	Inlet hydrogen/consumed hydrogen	1.5
F_H2_	Hydrogen flow rate	0~0.63 L/min
PWM	Pulse width modulation	15%
P_fan_	Fan power	18 W (V = 12 V, I = 1.5 A)
n_cell_	Cell numbers	1
RH_H2_	Hydrogen relative humidity	0%
RH_Air_	Air relative humidity	30∼45%
P_H2_	Hydrogen inlet pressure	50 kPa

## Data Availability

Not applicable.

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
