# Peer review of "Experimental Investigation on the Anode Flow Field Design for an Air-Cooled Open-Cathode Proton Exchange Membrane Fuel Cell"

_membranes, 2022, doi:10.3390/membranes12111069_

Round 1

Reviewer 1 Report

This article addresses the effects of the anode serpentine flow field parameters (nº of channels and channels width and depth) on the performance of an open-cathode PEM fuel cell.

The topic seems relevant at first, as there are very few studies on the flow field design of the anode of an open-cathode PEM fuel cells designed for portable applications such as UAVs. However, going over the paper one finds that the anode is operated at constant stoichiometry of 1.5 (according to table 3). As far as the reviewer’s knowledge, open-cathode PEM fuel cells for portable applications operate in dead-end anode mode (anode outlet closed with periodical purges to remove excess water and N2 that might accumulate in this electrode). That is referred by the authors in lines 92 and 93. Operating at constant stoichiometry in the anode decreases the relevance of the paper as the flow characteristics would not be much different from “more traditional” PEMFCs operation. It would be of greater relevance if the experiments were conducted with dead-end anode operation.

Specific comments/questions:

1-      At the end of the Introduction, the authors refer to be presenting a “different anode model”? what does it mean? Where is this new model? According to the reviewer’s knowledge, equations in section 2 are not new.

2-      I have difficulty understanding the relevance of section 2. It seems the only output is the calculation of pressure drop in the anode flow field for the different configurations tested. The results are somewhat intuitive. The pressure drop could also be measured in the experiments. The same could be said about the several mentions in the text about field synergy theory. The reviewer does not understand the relevance of mentioning it in the context of the present study. That can confuse the reader.

In essence, in the reviewer’s view the section 3. Results and Discussion section could be standing-alone in this paper.

3-      “otherwise” seems not being used correctly. Please check.

4-      Lines 344 and 372. The fact that the performance varies with the channel features also affects heat generation. Better performance, less heat generation thus lower temperature. Attribute the differences in temperature only to heat dissipation might not be correct. Please check.

5-      Line 350: There seems to be missing an “and” after “flow resistance decreases”

6-      Table 3: (1) how do you know the heat generation of the stack and is it always the same? (it varies with the performance); (2) P_H2 is inlet pressure, back pressure?

7-      Conclusions: (1) please explain the “novel flow field design”. The flow field designs tested are not new… (2) Please explain “combined with field synergy theory”.

The reviewer considers section 3. fairly well written and interesting, despite the issue of relevance explained above. However, section 2. and the several mentions of field synergy theory does not seem to add considerable value to the paper and make the article confusing and its aims different to understand for the reader.

Author Response

Thank you for your valuable suggestion. We have made corresponding modifications to all your suggestions. See the attachment for details.

Thank you for your approval of the research content of this article, and hope that our modifications can meet your expectations.

Reviewer 2 Report

The manuscript by Deng et al. reveals the important parameters of the flow channel structure design in PEMFC.

1.      In the introduction (page 2) I’d recommend focus on the results of various researchers, starting from the reference [19], rather than just mentioning the subjects of their studies.

2.      Please, specify the membrane (CCM) used in MEA (section 3.1). The preparation of MEA is very briefly described. How the channels of such design were fabricated?

3.      The abbreviation EIS is not presented at the first usage (line 248).

4.      The sentence at lines 264-265 leaves some doubts: “The cell with a 6-serpentine channel can achieve a 10.7% performance improvement at 0.8 A/cm2 compared with the cell with a 2-serpentine channel.” What does it mean – the improvement of 6-serpentine channels is 10.7% larger or achieves 10.7% from the level of 2-serpoentine channels, i.e. lower? Please, formulate it in a more clear way.

5.      The data In Figs. 3a, 5a and 7a should be related with the left and right axes.

6.      Conclusions require a few more discussion on the scientific results of the work, specific recommendations on the MEA design to achieve better performance, etc.

Author Response

(The authors gave the same response as above.)
